# Influence of Varied Load Assistance with Exoskeleton-Type Robotic Device on Gait Rehabilitation in Healthy Adult Men

**DOI:** 10.3390/ijerph19159713

**Published:** 2022-08-06

**Authors:** Toshiaki Tanaka, Ryo Matsumura, Takahiro Miura

**Affiliations:** 1The Research Center for Advanced Science and Technology, Institute of Gerontology, The University of Tokyo, Tokyo 113-8656, Japan; 2Department of Physical Therapy, Faculty of Health Sciences, Hokkaido University of Science, Sapporo 006-8585, Japan; 3National Institute of Advanced Industrial Science and Technology (AIST), Kashiwa 277-0882, Japan

**Keywords:** robotic rehabilitation, motion analysis, gait-assist devices

## Abstract

This study aimed to clarify how the power-assist function of the hybrid assistive limb (HAL^®^), an exoskeleton-type gait-assist device, affected the gait characteristics of patients who wear it, specifically focusing on the “misalignment” of the robot joints and landmarks with the corresponding body parts. Five healthy adult men were video-recorded wearing the HAL^®^ as they walked normally on a treadmill under seven conditions corresponding to the strengths and sites of robotic power assistance. For kinematic analysis, reflective markers were attached to the HAL^®^ and the wearer at key locations, and participants were recorded walking past a set of four video cameras for each condition. A motion analysis system was used for analysis. The walking motion was segmented into eight-phase gait cycles. Knee misalignment, or the relative offset in position of the HAL^®^/wearer knee joints, was calculated from kinematic data and joint angles. These values were compared with respect to two factors: assist level and gait phase. Statistical analysis consisted of parametric and nonparametric tests for comparing the values of misalignment of each gait phase, followed by multiple comparisons to confirm significant differences. The results showed that the knee misalignment was greatest in the pre-swing phase and was significantly lower overall in conditions with high levels of power assistance. The result of greater knee misalignment in the pre-swing phase may be attributed to the structural properties of the HAL^®^ lower-limb exoskeleton. This provides valuable insight regarding the walking stages that should be given special attention during the evaluation of synchrony between exoskeleton-type gait-assist robots and their wearers.

## 1. Introduction

The Medical Device and Healthcare Project is one of the six integrated projects sponsored by the Japanese government aimed at promoting research and development in the medical field for providing world-class healthcare, specifically focused on artificial intelligence (AI), and outlined in the early 2020 Cabinet Office publication, “Health and Healthcare Strategies”. The project’s stated aim is “to apply a fusion of AI/Internet of things, measurement, and robotics technologies to research and develop medical devices/systems for the advancement of diagnostics and treatments, medical devices strongly needed in clinical settings, and medical devices and healthcare for preventing disease and improving quality of life” [1]. Robots are being introduced into medicine and long-term care in Japan, and some rehabilitation robots have been eligible for coverage by national health insurance since April 2016. Furthermore, the 2020 revision of medical service fees introduced an addition (or credit) for devices that facilitate exercise in cases of rehabilitation services, which are counted as neurovascular rehabilitation-using eligible devices [2,3]. Research on the application of robots in rehabilitation (hereafter referred to as “robotic rehab”) has given rise to expectations regarding their future applicability in the fields of medicine and social welfare in recent years, and clinical research in this area has been gradually expanding in Japan. One area of research that has drawn special attention involves robots fitted directly to the human body to provide a power-assist function. Studies have been conducted using power-assist robots to support long-term care, rehabilitation, and daily activities.

The role of robotic rehab for post-stroke gait disturbance, which mainly involves externally supporting the movement of the hip and knee joints of paralyzed lower extremities, has expanded. In rehabilitation training, robots have been used to provide patients with external assistance for optimal strength (the minimum assistance needed to accomplish a task) and sufficient precise repetition to aid motor learning, which is anticipated to improve the efficiency of rehabilitation training. Examples of training support robots are Lokomat© from Hocoma, Gait Trainer© from Reha-Stim Medtec, and HAL© from CYBERDYNE Inc. (Tsukuba, Japan) [4,5,6,7]. Presently, the main advantages of introducing robots are that they can be used for training for repetitive movements that are difficult for patients and as a basis for enabling PTs to concentrate on qualitative training. Robot therapy is actively being researched from the viewpoint of motor learning and has a clinical application in rehabilitation [8,9]. In particular, several recent studies have reported evidence for the feasibility, efficacy, and safety of the HAL^®^, when utilized in rehabilitation interventions for patients with hemiplegic stroke [10,11]. Two crucial components of HAL^®^-based rehabilitation are the intentions of the training, i.e., deciding what kinds of movements to support based on the opinions of the therapist and patient, and the configuration of the control parameters. Today, the burden of guiding patient movements largely falls on the therapist’s expertise, while the suit’s control parameters are equally important to the HAL^®^’s effective application. However, few teams have investigated the movements in detail, making it unclear how improper device–wearer fit can affect its motion assistance function. Gait-support exoskeleton robots, typified by the HAL^®^ Robot Suit, reinforce the direction of the wearer’s movements by complementing their joint motion with their frame and rotation axes. However, since the robotic joints have fewer degrees of freedom than their physiological equivalents, the exoskeleton’s structure constrains the wearer’s movements, making it unclear whether the device adequately supports walking movements.

Rehabilitation medicine could be improved immensely by treatment and training equipment that effectively incorporate robotic technology [12,13]. Several groups have reported positive outcomes for gait-support robots in walking rehabilitation; however, most of these examined only changes in gait parameters after device use or robot-assisted training [10,14,15]. Few studies have evaluated gait parameters while patients are wearing such devices. Thus, this study aimed to clarify how the power-assist function of the hybrid assistive limb robot suit, an exoskeleton-type gait-assist device, affects the movement of joints during gait. This study specifically focused on identifying any misalignment of robot joints or landmarks with their corresponding body parts. The purpose was to show the effects of robotic assistance devices on their wearers, along with related technical challenges, and to provide valuable insights concerning gait-support robots and their effective utilization in rehabilitation practice. It was hypothesized that the stance and swing phases would be associated with the greatest misalignment between the robotic and wearer’s knee joints.

## 2. Materials and Methods

### 2.1. Participants

Five healthy adult men (age 38.4 ± 8.0 years, height 170.4 ± 5.0 cm, and weight 67.2 ± 3.3 kg) participated in this study. Participants were required to be ≥20 years of age and able to independently provide written informed consent. To wear the HAL^®^, patients needed to be 150–185 cm in height, 40–80 kg in weight, and have a suitable lower-body profile (femur/calf length, waistline) for the device. Past medical history of any condition that could affect the gait was deemed exclusionary. All participants had a normal gait. Normal gait is characterized as symmetrical; this definition is supported by several authors who did not observe differences in ground reaction forces between the two legs during walking [16].

All participants received verbal and written explanations of the study before providing their written informed consent, with an assurance that their participation would be strictly voluntary, that their non-participation would not result in any disadvantage, and that their personal information would be protected. This study was conducted with the approval of the institutional review board of The University of Tokyo (approval review No. 20-210).

### 2.2. Device Used: A Robot-Assisted Gait Training Device

#### 2.2.1. Key Principles

The device used in this study was the HAL^®^ robot suit, an exoskeleton-type gait-assist robot that supports wearers’ body and joint movements, controlling them in real time based on feedback from sensors for myoelectric potentials, foot pressure, joint angles, and other biometric data. Power assistance is a key operating principle; sensor feedback and bioelectric information are relayed to actuator units positioned at lower-body joints, allowing the suit to support users’ voluntary movements by synchronizing with their intentions [6,17].

The mode and assist settings for the HAL^®^ are specified below:①Voluntary control mode: Myoelectric potentials (electromyography [EMG] activity) from the flexor and extensor muscles of the hip and knee joint are sensed by electrodes, and the center of pressure at the foot is sensed via specialized shoes. An assist level is then selected, and joint movement is controlled at the calculated “assist torque (Nm)”.②Impedance control mode: Weight-bearing and joint movement are smoothly controlled in synchrony with voluntary control mode and without assistance.③Assist level: The settings for the hip and knee joint actuators can each be adjusted across a range of 0–20 levels. An assistance level can indicate an assist torque value if a myoelectric potential value is described. The assist torque is defined as an assist level multiplied by the myoelectric potential value.

#### 2.2.2. Robotic Lower-Limb Exoskeletons and Device–Wearer Fit

The HAL^®^’s exoskeleton essentially consists of a lower-body frame (including waist and legs) with specialized shoes and actuator units at the hips and knees. It is critical to ensure that the HAL^®^ fits the wearer, so the suit properly synchronizes with their movements; when the device is fastened to a patient during rehabilitation, therapists should repeatedly adjust and evaluate device–wearer fit. Proper alignment between the robotic and physiological joints is especially important at the knee, but achieving this correctly is a challenge, even for the clinical lower-limb prostheses normally used for rehabilitation. Since poor fit can cause device slippage and even wearer injury in dynamic gait, ensuring the knee joints are properly aligned is a challenge that deserves special attention.

### 2.3. Trial Conditions

Participants were video recorded by four digital video cameras while wearing the HAL^®^ and walking on a treadmill at a speed of 1.5 km/h under various conditions (Figure 1). Captured video data were imported to a personal computer at a sampling frequency of 60 Hz using Frame-DIASV software (DKH Co., Ltd., Tokyo, Japan). Participants walked with seven different combinations of active joints and power assistance (Table 1). However, the HAL^®^ was active only on the participant’s left leg, while the right leg of the HAL^®^ was put in the impedance control mode without assistance. This experiment used just three of the 20 assist levels available when using the HAL^®^: Level 0 (no assistance), Level 1, and Level 3 (minor assistance). In this study, the mean assist torque of all participants was approximately 1 Nm for Level 1 and 3 Nm for Level 3. The combinations of assist levels that were tested are shown in Table 1: for example, “HIP1KNEE1” signifies Level 1 assistance at both the hip and knee joints. Conditions were recorded for 60 s each and separated by a rest period; a sequence of seven conditions was considered one run.

### 2.4. Reflective Markers

For kinematic analysis, reflective markers were attached to the left side of the HAL^®^ and the wearer, at a total of six locations: the hip, knee, and ankle joints of the HAL^®^, and the wearer’s knee (lateral aspect), ankle (lateral malleolus), and foot (fifth metatarsal head), as shown in Figure 2.

### 2.5. Analysis Methodology

#### 2.5.1. The Gait Cycle

The gait was analyzed according to the Rancho Los Amigos system, a functional classification consisting of eight phases. This system includes a major division between the stance phases, in which the heel is planted on the ground, and the swing phases, in which the foot is swung forward [18]. One gait cycle was defined as the full sequence of motion between two heel–ground contacts of the same leg. The eight-phase cycle is summarized in Figure 3.

#### 2.5.2. Device–Wearer Misalignment

First, a sequence of five full cycles of steady-state walking was extracted from each 60-s recording. To evaluate how the motions with HAL^®^ deviated from the user’s motion during gait training, the distance between two markers—the HAL^®^ knee joint and the wearer’s knee—was calculated in a 3D coordinate system. This value (misalignment) was calculated using the Frame-DIAS 3D motion analysis system (sampling frequency: 60 Hz).

Knee misalignment was calculated separately for each phase, relative to initial contact, as the absolute difference (mm) in the distance between the HAL^®^ knee and body markers in the phase of interest, minus the distance between the same markers at initial contact (no difference = 0 mm). Peak knee misalignment (i.e., the maximum misalignment observed) was calculated in each phase, then averaged across the five runs (participants) for phase-wise comparisons. Moreover, the angle of the knee joint was analyzed in the same five cycles. To evaluate how well the robotic joints of the HAL^®^ followed the wearers’ motion during gait training, the knee joint angles formed by the HAL^®^ and the body markers in the sagittal plane were calculated according to the definitions provided in Table 2. Since the robot suit surrounded the hip joint and foot of the body, these joint motions could not be measured. Joint angles were calculated using the Frame-DIAS 3D motion analysis system. This analysis examined the HAL^®^ and the body knee angles, recorded at the same time point as the peak knee misalignment, comparing the mean of the five runs recorded for each condition (i.e., one per participant).

### 2.6. Statistical Methodology

Statistical tests were performed to identify the differences between conditions across the five healthy adults. Peak misalignment (ankle, knee) and knee angle (HAL^®^, wearer) were compared by two factors: power-assist level and gait phase. When thus grouped, each variable’s data were shown to be non-normally distributed by the Shapiro–Wilk test (all *p* < 0.05). Further, homoscedasticity could be assumed, because Bartlett’s test for equal variances was not significant (*p* > 0.05). Based on the above preliminary statistical results, these data led us to apply two-factor aligned rank transform (ART) as a nonparametric analysis of variance [19]. When a factor was found to have a significant main effect and/or interaction, its levels were compared using least square means multiple comparisons (LSM-MC) [20,21]. Statistical analysis was performed using R version 4.12. The significance level was set at 5% for all analyses.

## 3. Results

### 3.1. Precision of Misalignment Data

To verify the precision of the recording technique, variability was examined in two of the input terms in the equation for calculating misalignment: namely, the relative distance between the HAL^®^ and a body marker at initial contact A and the same at each gait phase B. The dispersion of the collected data was analyzed using the coefficient of variation (CV), a measure representing dispersion within a data sample. The CV (%) is equal to the standard deviation of a sample divided by its mean, multiplied by 100, and can be defined as a sample’s mean variability [22]. The CV increases with increasing variation and decreases with decreasing variation [23]; values below 10% are typically interpreted as reflecting highly reproducible data [24]. The CV was calculated for samples A and B from the knee and ankle markers based on the average of five participants. The mean CV was <5% at the knee, confirming that every sample had been precisely and reliably obtained (Table 3).

### 3.2. Peak Knee Misalignment

The figures below compare the peak knee misalignment by the two factors of interest: gait phase (Figure 4) and assist condition (Figure 5). Two-factor ART ANOVA (gait phase × assist condition) showed each factor to have a significant main effect on the peak knee misalignment [F (6, 196) = 9.60, *p* < 0.001, η^2^ = 0.076; F (6, 196) = 2.76, *p* = 0.013, η^2^ = 0.026], but there was no significant interaction with each other [F (36, 196) = 0.479, *p* = 0.995, η^2^ = 0.027]. These results signified the presence of differences in peak knee misalignment attributable to the gait phase, distinct from other differences attributable to the power assistance.

Next, LSM-MC was performed as a secondary test to characterize the main effect of the gait phase in greater detail. The peak knee misalignment was significantly larger in the pre-swing phase than all others compared: loading response [t (4) = 6.33, *p* < 0.001, r = 0.95], midstance [t (4) = 3.27, *p* = 0.021, r = 0.85], terminal stance [t (4) = 3.89, *p* = 0.003, r = 0.89], initial swing [t (4) = 3.52, *p* = 0.009 r = 0.87], midswing [t (4) = 4.02, *p* = 0.002, r = 0.90], and terminal swing [t (4) = 6.57, *p* < 0.001, r = 0.96]. In addition, the peak knee misalignment was significantly smaller in the terminal swing phase compared with the midstance phase [t (4) = 3.30, r = 0.86, *p* = 0.019] or initial swing phase [t (4) = 3.05, r = 0.84, *p* = 0.041], and in the loading response compared with the midswing phase [t (4) = 3.06, r = 0.84, *p* = 0.040]. Next, LSM-MC was performed as a secondary test to characterize the main effect of the assist level in greater detail. The peak knee misalignment was significantly greater in Hip1 than Hip3Knee3 [t (4) = 3.09, *p* = 0.036, r = 0.84].

## 4. Discussion

In evaluating the effects of the gait phase and the assist level on knee alignment, no statistical interaction was found between these factors. The knee moves through two sets of flexion and extension in a normal gait cycle: the first occurs from initial loading to midstance, and the second occurs from pre-swing to the initial swing [18,25]. A few studies have suggested that some misalignments between the user’s anatomical and exoskeleton joints can cause undesired interaction forces, which in turn reduce comfort and safety [26,27,28]. Misalignments have been discussed as a potential cause for lower limb fractures for a powered exoskeleton use [29].

The experimental data for the knee showed the effect of misalignment in the pre-swing phase to be greater than that in all other phases of the gait cycle. One reason was that in the pre-swing phase, the hip joint was in the flexed position, mainly due to the activity of the hip flexor muscles, after which the positions of the knee joint flexion and the ankle joint plantar flexion were maintained simultaneously. Essentially, the pre-swing phase magnifies the knee flexion begun in the terminal stance. Therefore, the stronger the assistive control for flexion of the hip joint of the HAL^®^, the more difficult it may be for the human knee joint to follow.

Further, the portion of the knee joint between the HAL^®^ and the body could be mechanically unstable. Structural aspects of the HAL^®^ exoskeleton may offer some insight into these motions. When a patient is equipped with the device, their lower torso and pelvis are strapped to the corresponding parts of the frame by belts. Belts also fasten their thigh and calf to the respective cuffs, and the patient’s feet are placed in specialized shoes connected to the outer frame. The device does not mechanically immobilize or restrain the wearer’s leg at the knee. Therefore, this structural aspect of the lower HAL^®^ exoskeleton may have been related to the greater knee misalignment observed around the beginning of the leg’s swing forward, caused by the greater forces delivered by the suit in the direction of the knee flexion. The power output—or assist torque—from the HAL^®^ primarily depends on sensor readings of muscle activity. In this study, the hip flexor activity in the pre-swing phase could have exacerbated knee misalignment by triggering greater assist torque at the robotic hip joint. The patterns of muscle activity of the lower limbs during walking were not analyzed in this study; thus, to clarify the effects of robotic power assistance in different gait phases in greater detail, future work will need to incorporate EMG data and analyze impedance and other bioelectric signals that can be tracked by the HAL^®^.

Regarding the factor of the assist condition, it was initially hypothesized that the knee misalignment between HAL^®^ and the wearer would be greater under conditions of strong assistance and worse at the knee than at the hip joints. The data showed that the knee misalignment was smaller in HIP3 KNEE3 (assist Level 3 for the knee and hip joint) than in HIP1 (assist Level 1 for the hip joint), indicating greater stability resulting from the addition of the knee actuator and stronger loads overall. The strength of the assist torque primarily depends on sensor readings of muscle activity. The significantly lower misalignment was observed in the HIP3 KNEE3 condition involving moderate torque at both the hip and knee compared with the HIP1 condition involving mild assistance only at the hip. This result was attributed to the stabilization of the wearer’s joint motion caused by greater output at the hip joint and additional control at the knee. In this study, the effect of a gait-support robot suit on the wearer’s walking behavior varied by gait phase and power-assist level. For example, if the gait disorder is mild, the misalignment between the robot suit and the human body during walking may increase if the “mild” assist control mode is used. Therefore, a new method may be needed for fixing the alignment between the suit and the body at the joint site during gait rehabilitation using a robot.

This study had some limitations. First, there was a small sample size of five participants, which can be attributed to the restrictions imposed by the COVID-19 pandemic. A larger sample size is recommended for future studies. Second, the study used a low assist level of the HAL for those who needed slight support. For example, patients with stroke may not be able to have voluntary muscle activity if they use a higher level of assistance because of their strong dependence on the HAL. A stroke patient with strong spasticity will need a high assist level with the HAL to improve gait disturbance. In addition, older people with pre-frailty may need low assistance for their gait clearance. Data on misalignment at versatile assist levels should be obtained in the future. Third, this study focused on the knee joint; possible misalignment of the hip and ankle should be measured and analyzed further. Fourth, the walking speed was set to 1.5 km/h (0.4 m/s) because of the assumption that frail older people and disabled people who need the HAL walk slowly. Ideally, the study would be conducted at various speeds to analyze the different misalignments. Finally, there were no three-dimensional (3D) misalignment and EMG data. Further investigations using multiple markers by 3D motion analysis and EMG are required.

## 5. Conclusions

This study sought to clarify the effects of power assistance in gait-support devices based on kinematic data captured from participants who wore the robot HAL^®^ suit, a robotic lower-limb exoskeleton suit, while walking. The gait phase analysis confirmed the misalignment between the robotic and biological knee joints during the gait cycle. Furthermore, greater power assistance at the hip and knee reduced the misalignment more than power assistance at the hip only. Further studies are required to investigate the need for an assist level setting based on each gait phase and to consider a new method for fixing the alignment between the suit and the body at the joint site during gait rehabilitation using robot therapy.

## Figures and Tables

**Figure 1 ijerph-19-09713-f001:**
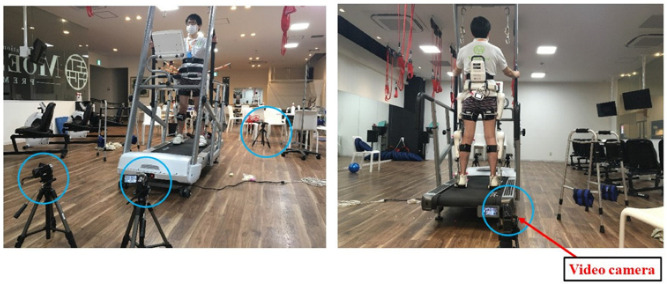
Experiment: walking on a treadmill wearing a robot suit.

**Figure 2 ijerph-19-09713-f002:**
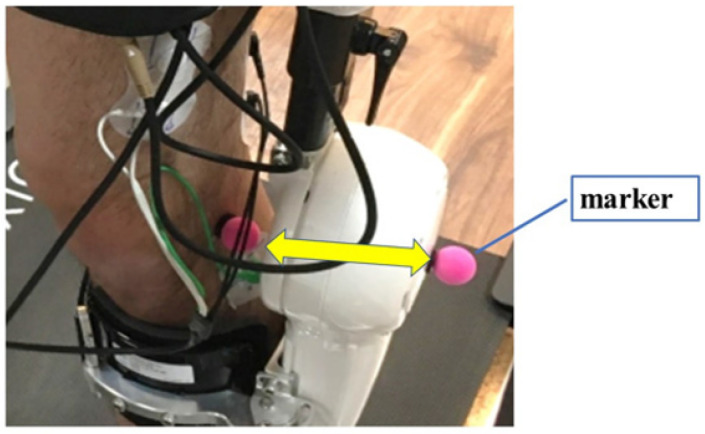
Mounting position of the reflective markers of the HAL^®^ and the body at the knee joint for motion analysis.

**Figure 3 ijerph-19-09713-f003:**
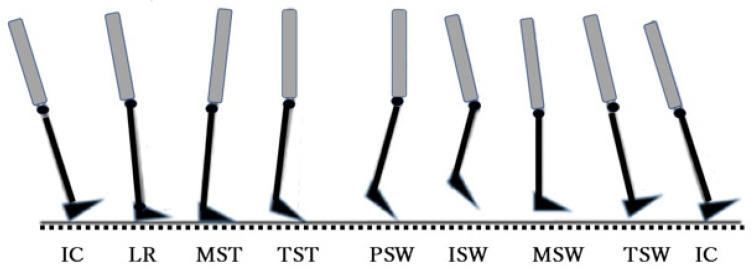
Gait analysis: the eight-phase gait cycle. IC: Initial Contact, LR: Loading Response, MST: Midstance, TST: Terminal Stance, PSW: Pre-Swing, ISW: Initial Swing, MSW: Midswing, TSW: Terminal Swing.

**Figure 4 ijerph-19-09713-f004:**
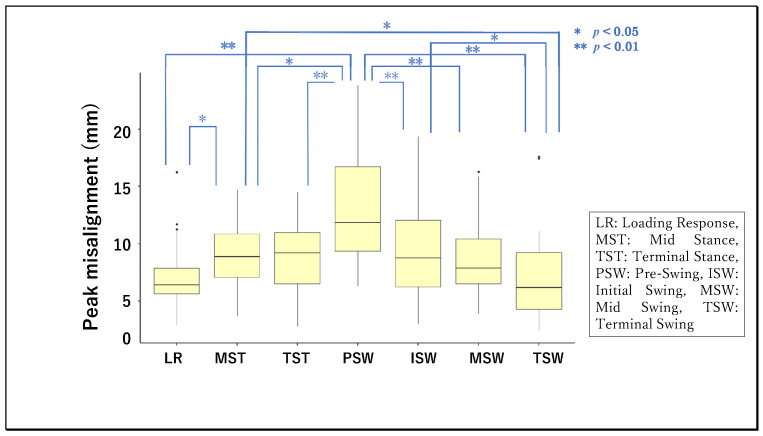
Mean values of peak misalignment of knee for each gait phase.

**Figure 5 ijerph-19-09713-f005:**
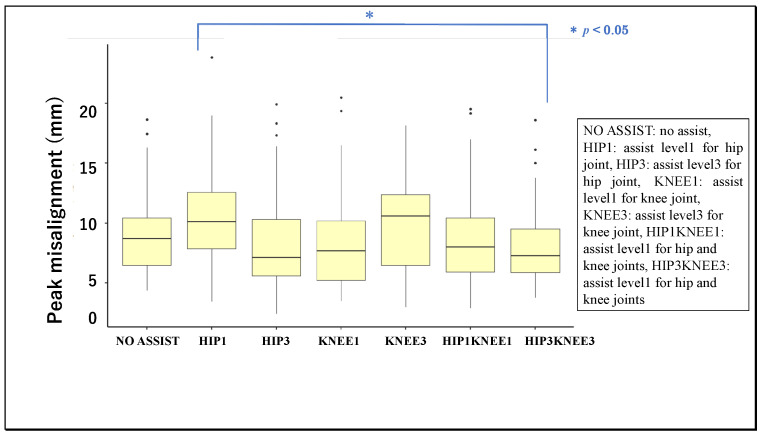
Mean values of peak misalignment of knee for different assist levels.

**Table 1 ijerph-19-09713-t001:** The conditions of assist level of the hip and knee joints for the HAL^®^.

Assist Condition	Assist Level (0, 1, 3)	
	**HIP**	**KNEE**
NO ASSIST	0	0
HIP1	1	0
HIP3	3	0
KNEE1	0	1
KNEE3	0	3
HIP1KNEE1	1	1
HIP3KNEE3	3	3

NO ASSIST: no assist, HIP1: assist level 1 for hip joint, HIP3: assist level 3 for hip joint, KNEE1: assist level 1 for knee joint, KNEE3: assist level 3 for knee joint, HIP1KNEE1: assist level 1 for hip and knee joints, HIP3KNEE3: assist level 3 for hip and knee joints.

**Table 2 ijerph-19-09713-t002:** The definition of angles for the hip, knee, and ankle joints of the HAL^®^ and the body.

HAL^®^ knee joint angle: HAL^®^ hip joint—HAL^®^ knee joint—HAL^®^ ankle joint
Body knee joint angle: HAL^®^ hip joint—body lateral aspect of knee—body lateral malleolus
HAL^®^ ankle joint angle: HAL^®^ knee joint—HAL^®^ ankle joint—HAL^®^ 5th metatarsal head
Body ankle joint angle: body lateral aspect of knee—body lateral malleolus—HAL^®^ 5th metatarsal head

**Table 3 ijerph-19-09713-t003:** Mean coefficient of variation (CV) of misalignment data in the gait phase.

	IC	LR	MST	TST	PSW	ISW	MSW	TSW
CV (%)	2.6	2.9	3.3	3.4	4.1	2.9	2.7	2.7

IC: Initial Contact, LR: Loading Response, MST: Mid Stance, TST: Terminal Stance, PSW: Pre-Swing, ISW: Initial Swing, MSW: Mid Swing, TSW: Terminal Swing.

## Data Availability

The datasets analyzed during this study are available from the corresponding author upon reasonable request.

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
