# Peer review of "Influence of Varied Load Assistance with Exoskeleton-Type Robotic Device on Gait Rehabilitation in Healthy Adult Men"

_ijerph, 2022, doi:10.3390/ijerph19159713_

Round 1

Reviewer 1 Report

See pdf for comments.

Author Response

We sincerely appreciate all valuable comments and suggestions, which helped us to improve the quality of the manuscript. The responses to the Reviewers’ comments are described in our uploaded file letter.

Reviewer 2 Report

The article presents an interesting activity in the problem of misalignments among the centers of rotation of exoskeleton and centers of joints of a patient. Some improvements have to be reported.

-       Correct numbers of references in the list of references. These numbers do not correspond with numbers in the article text.

-       Table 1: the values of assist levels do not correspond to assist conditions. Please correct.

-       Table 1 caption:  … HIP3KNEE3: assist level 3 (NOT 1).

-       On page 4, lines 153-154 the values of 1 Nm and 3 Nm are considered. For a person of about 67 kg, the hip torque is about 50-70 Nm and for knee about 50 Nm. Why so low levels are considered?

-       In the discussions of results the authors could improve their conclusions, considering the different effects of the patient weight on a leg in different conditions: in stance phase the patient weight acts on a leg that is in contact with the terrain; in swing phase the patient weight do not acts on the leg. If possible the authors should say something about these effects.

Author Response

(The authors gave the same response as above.)

Reviewer 3 Report

I appreciate the opportunity to review this manuscript. I aim to provide a fair and kind review and hope that my comments will be accepted in that manner.

The area of exoskeletons within rehabilitation is an interesting and emerging field that is now receiving much attention. Please find below comments and suggestions based on the manuscript reviewed.

-The manuscript generally requires proof reading and should focus more directly on the results and findings presented only in the results section of this paper

-Add the study population to title-able-bodied adult males

-N=5 seems low? Can conclusions be drawn from this? Was a power calculation completed? This should be added to a limitations section

-Clarification is required in the abstract regarding the population-able bodied? Stroke rehab? Neurological impairment? Etc

-Line 23 remove ‘we’ and all first-person references throughout the manuscript

-Is the first paragraph (lines 33-39) of the introduction required? Seems very abstract- remove and include one line introducing the area of robotics in clinical/rehab settings

-Intro paragraph requires reformatting- some paragraphs currently exist with 2 sentences!! Combine one topic/notion in one paragraph

-add in line 66 exobionics and reference Duddy et al 2021 and 2022 papers

-font sizes used on all table and figures needs revised-some appear too large

- Figure 3 should read ‘Gait’

-Hal is missing ® often in document e.g., in headings and figures, this should be consistent

-line 237 remove ‘a’ from ‘p = .041a’?

-line 256 remove double space

-line 265 ‘however’ should not be the end of a sentence?

-The manuscript requires proof-reading

-line 273/4 is the EMG abbreviation alone enough here as you provide the abbreviation in abstract?

-can you draw these conclusions using able bodied individuals? Surely this is a limitation of the study and should be highlighted as such

-the assisted mode may only be helpful within impaired individuals? It may actually (and should) distort the alignment of individualised walking gait within able-bodied adults as their gait will naturally be individualised and have many effectors e.g. muscle strength, limb length etc etc

-what are the limitations of this study? There are numerous- this should be a section

-conclusions should be concise and redrafted based on the specific study conclusions.

-line 28 of the abstract states …’our findings on electromyography (EMG)’ suggesting these will be provided in the manuscript but these do not appear until the authors suggest in the discussion and conclusion section that this work is required!!? Please focus on the results of the findings presented in this manuscript instead of wording the findings to suggest there is need for presumably your next paper?!!

-The reference list is substantially inappropriate in its current form and requires substantial amendments 

Author Response

(The authors gave the same response as above.)

Round 2

Reviewer 1 Report

Thanks for the response. There are some additional comments and questions that should to be addressed:

1) There are some minor flow and grammar issues, please proofread it again

2) Line 100: you need to define or reference what you mean by "normal gait".

3) Discussion Section - You didn't discuss the potential impact of the misalignment seen on your results on the joint biomechanics of the individual. i.e. why should the reader care about misalignment? how will this affect people's joint health? will it increase the risk of injury or osteoarthritis? You could use literature data for this.

4) Figure 5: please put your legend some where on the chart or put you caption under the legend because it is a bit confusion the way it is displayed

5) There should be other misalignment studies in the literature. You should give a comparison to the results you found in your study in your discussion. You do not need to have other robots to do this comparison. You could use literature data for this.

Author Response

Dear reviewers,

We sincerely appreciate all of the valuable comments and suggestions we received from you: they have helped us improve the quality of our manuscript. Our point-by-point response to each reviewer’s comment is outlined below. Additionally, the necessary revisions, as suggested by the reviewers, are indicated as text in blue-colored font within the revised manuscript. We have also had the manuscript proofread by Editage, a professional English editing service (https://www.editage.com/).

Reviewer 2 Report

The authors improved the paper with new comments. The pare may be accepted in the present form.

Author Response

(The authors gave the same response as above.)
